Comparative analysis of whole flower transcriptomes in the Zingiberales

Almeida Ana Maria R. ana.almeida@csueastbay.edu 1
Piñeyro-Nelson Alma 2
Yockteng Roxana B. 3 5
Specht Chelsea D. 4
1 Department of Biological Sciences, California State University, Hayward , Hayward , CA , United States of America
2 Department of Food and Animal Production, Autonomous Metropolitan University, Xochimilco , Mexico City , DF , Mexico
3 Centro de Investigaciones Tibaitatá, Corporación Colombiana de Investigación Agropecuaria (AGROSAVIA) , Tibaitatá , Colombia
4 School of Integrative Plant Sciences, Section of Plant Biology and the L.H. Bailey Hortorium, Cornell University , Ithaca , NY , United States of America
5 Institut de Systématique, Evolution, Biodiversité-UMR-CNRS, National Museum of Natural History , Paris , France
Van de Peer Yves
Electronic publication date: 2018 Aug 24
Publication date: 2018
Volume: 6
Electronic Location ID: e5490
Received 2017 Jul 6; Accepted 2018 Jul 30
Copyright: ©2018 Almeida et al.
Copyright year: 2018
Copyright holder: Almeida et al.
License: This is an open access article distributed under the terms of the Creative Commons Attribution License, which permits unrestricted use, distribution, reproduction and adaptation in any medium and for any purpose provided that it is properly attributed. For attribution, the original author(s), title, publication source (PeerJ) and either DOI or URL of the article must be cited.
License URL: https://creativecommons.org/licenses/by/4.0/

Keywords: Ginger transcriptomes, Floral evolution, Floral evo-devo, Monocot flower, Floral transcriptomes

Funding: NSF IOS 0845641 DEB 0816661 DEB 1257701 INCT IN-TREE, 465767/2014-1 CAPES BJT 069/2013 UC-MEXUS-CONACYT postdoctoral scholarship This work was supported by UC Berkeley College of Natural Resources, NSF IOS 0845641, DEB 0816661 and DEB 1257701 to Chelsea Specht, and INCT (IN-TREE, 465767/2014-1) to Ana Almeida. Ana Almeida was supported by a Bolsa Jovens Talentos (CAPES, Brazil, BJT 069/2013). Alma Piñeyro-Nelson was supported by a UC-MEXUS-CONACYT postdoctoral scholarship. The funders had no role in study design, data collection and analysis, decision to publish, or preparation of the manuscript.

==============================
The advancement of next generation sequencing technologies (NGS) has revolutionized our ability to generate large quantities of data at a genomic scale. Despite great challenges, these new sequencing technologies have empowered scientists to explore various relevant biological questions on non-model organisms, even in the absence of a complete sequenced reference genome. Here, we analyzed whole flower transcriptome libraries from exemplar species across the monocot order Zingiberales, using a comparative approach in order to gain insight into the evolution of the molecular mechanisms underlying flower development in the group. We identified 4,153 coding genes shared by all floral transcriptomes analyzed, and 1,748 genes that are only retrieved in the Zingiberales. We also identified 666 genes that are unique to the ginger lineage, and 2,001 that are only found in the banana group, while in the outgroup species Dichorisandra thyrsiflora J.C. Mikan (Commelinaceae) we retrieved 2,686 unique genes. It is possible that some of these genes underlie lineage-specific molecular mechanisms of floral diversification. We further discuss the nature of these lineage-specific datasets, emphasizing conserved and unique molecular processes with special emphasis in the Zingiberales. We also briefly discuss the strengths and shortcomings of de novo assembly for the study of developmental processes across divergent taxa from a particular order. Although this comparison is based exclusively on coding genes, with particular emphasis in transcription factors, we believe that the careful study of other regulatory mechanisms, such as non-coding RNAs, might reveal new levels of complexity, which were not explored in this work.

Introduction

Next-generation sequencing technologies have been instrumental in allowing for the rapid generation of large quantities of transcriptomic data, previously unavailable for the majority of non-model organisms. In parallel to refinements of the sequencing technologies, several bioinformatics pipelines have been put forward allowing for the de novo assembly of transcriptomes from organisms for which there is not a fully sequenced and annotated genome (‘reference genome’ e.g., Wit et al., 2012; Chiara, Horner & Spada, 2013; Singhal, 2013; Unamba, Nag & Sharma, 2015). Although long predicted as a revolutionary tool (Wang, Gerstein & Snyder, 2009), RNA-Seq approaches enabling the comparative quantification of gene expression during organismal development have recently gained wide use across a diversity of organisms representing unique developmental and physiological processes. These advances have enabled the identification of candidate genes involved in a variety of processes ranging from flower color (e.g., pigment biosynthesis in Camellia reticulata (Yao et al., 2016); color polymorphism in Silene littorea Brot (Casimiro-Soriguer et al., 2016)) to characterization of biosynthetic pathways (e.g., glucosinolate and phytochelatin pathways in Sinapsis alba L. (Zhang et al., 2016); flavonoid and stilbenoids pathways in Gnetum parvifolium (Warb.) W.C.Cheng (Deng et al., 2016)). NGS approaches have also been used to study plant architecture (González-Plaza et al., 2016) as well as specific aspects of reproductive development (Hollender et al., 2014).

Chanderbali and colleagues (2009; 2010) pioneered the use of next-generation sequencing technologies to study the comparative evolution of floral development across angiosperms. Their choice of plant species included representatives of main angiosperm lineages (i.e., water lily, avocado, California poppy, and Arabidopsis), as well as a non-angiosperm seed plant (cycad), which allowed the authors to obtain insights into the molecular mechanisms underlying the evolution and diversification of the flower (Chanderbali et al., 2010). While there was deep conservation in the genetic programs specifying floral organ identities, further confirmed by the careful study of 18 angiosperm genomes (Davila-Velderrain, Servin-Marquez & Alvarez-Buylla, 2013), it was also possible to identify distinct transcriptional programs characterizing more recently derived plant lineages (Chanderbali et al., 2010). Thus, one can hypothesize that these distinct transcriptional programs are likely involved with mechanisms of diversification in floral shape, especially in closely related species.

In order to gain insight into the genetic basis of floral morphological variation, we present a comparative transcriptomic analysis of several species within the angiosperm order Zingiberales. The Zingiberales is a lineage of tropical and subtropical monocots comprising eight families. The order includes economically important species such as culinary ginger (Z. officinale Roscoe), turmeric (Curcuma longa L.), and banana (Musa acuminata Colla), as well as popular ornamentals, such as Canna indica L., bird-of-paradise (Strelitzia reginae Banks), spiral gingers (Costus spp.), and heliconias (Heliconia spp.). A recent phylogenetic analysis (Sass et al., 2016) supports the placement of Musaceae as sister to all other lineages followed by a monophyletic clade comprising Heliconiaceae, Strelitziaceae and Lowiaceae. Together, these four families are referred to as the “banana lineages” and form a basal paraphyly with respect to the derived monophyletic ginger clade (Cannaceae, Marantaceae, Costaceae, and Zingiberaceae = “ginger clade”) (Fig. 1A). Flower morphology in the Zingiberales varies dramatically, and one of the main floral transitions in the order is related to the androecial whorl. Throughout the evolution of the Zingiberales, the number of fertile stamens is drastically reduced from 5–6 fertile stamens in the banana lineages to one or 12 fertile stamen in the ginger clade. This reduction in fertile stamen number is inversely correlated to an increase in petaloidy, in which the infertile androecial members laminarize (flatten) and develop into petal-like organs (Almeida, Yockteng & Specht, 2015) (Fig. 1B).

Figure 1 Evolution of floral morphology in the Zingiberales.

(A) Most recent Zingiberales phylogeny (modified from Sass et al. (2016)). Zingiberales families are divided into the banana group, a paraphyletic assembly of early branching lineages, and the ginger clade. The asterix (*) marks the evolution of increased petaloidy and reduced number of fertile stamens as shared characteristics of the ginger clade. (B) M. basjoo flower and floral organs. Calix and corolla members are mostly fused into what is called the floral tube, with the exception of a single corolla member, the free petal. As a representative of the androecial constitution of the banana group, M. basjoo has five filamentous fertile stamens. M. basjoo gynoecium is also representative of most species in the banana group. (C) Canna sp. flower and floral organs. Species in the ginger clade usually exhibit inconspicuous and sepal-like calix and corolla, while infertile androecial members (staminodes) become laminar and petaloid. Species in the Zingiberaceae and Costaceae families bear a single fertile stamen, while species in the Cannaceae and Marantaceae families only develop 1/2 a fertile stamen. Furthermore, in Canna sp. the gynoecium is also laminarized to some extent. ft, floral tube; fp, free petal; se, sepals; pe, petals; st, stamen; th, theca; std, staminodes; gy, gynoecium (Photos by Ana Almeida).

Several gene and gene networks have been hypothesized as underlying the molecular mechanisms of Zingiberales floral developmental evolution (Bartlett & Specht, 2010; Yockteng et al., 2013a; Almeida et al., 2014; Almeida et al., 2015). However intriguing, these studies are limited to candidate-gene or candidate-process approaches. In this study, we present an analysis of whole flower transcriptomes of several species spanning the Zingiberales order, as well as of a closely related Commelinaceae species. We focus our comparative analysis on coding regions, with particular attention to transcription factors. This broad approach aims at avoiding the pitfalls of targeted candidate-based methodologies, and can potentially illuminate lineage-specific mechanisms of floral development linked to evolution and diversification in form and function. We also highlight the advancements and challenges of comparative transcriptome-based approaches for the study of developmental evolution.

Methods

Plant Material and RNA extractions

Whole developing flowers of Costus spicatus, Zingiber officinale, Calathea zebrina (Sims) Lindl., Canna sp., Orchidantha fimbriata Holttum, M. basjoo Siebold & Zucc., and Dichorisandra thyrsiflora were collected at the UC Berkeley Botanical Garden, Oxford Track Greenhouse, and UC Davis Greenhouse (Table 1). Whole young floral buds were collected and immediately flash frozen in liquid nitrogen. Flower and/or inflorescence size and morphology vary widely within the Zingiberales, and uniform developmental stages have not yet been established for the different lineages. In all cases, young inflorescences were dissected as much as possible and the youngest discernable floral buds were collected.

Table 1 Species used in this study, collection location and accession numbers.

Species	Location	Accession	
Dichorisandra thyrsiflora	UC Davis Greenhouse	B81.521	
Musa basjoo	UC Botanical Garden	89.0873	
Orchidantha fimbriata	Oxford Track Greenhouse (UC Berkeley)	194.656	
Canna sp.	Oxford Track Greenhouse (UC Berkeley)	KT795161	
Calathea zebrina	UC Botanical Garden	90.1656	
Zingiber officinale	Oxford Track Greenhouse (UC Berkeley)	KT795282	
Costus spicatus	Oxford Track Greenhouse (UC Berkeley)	KT795282	

Frozen floral buds were stored in −80 °C for up to two days before RNA extraction. Total RNA was extracted from floral material using Plant RNA Extraction Reagent (Invitrogen, Carlsbad, CA, USA), according to Yockteng et al. (2013b). RNA was stored at −80 °C until further use.

Library Preparation and sequencing

cDNA libraries for sequencing on the Illumina platform were prepared using the TruSeq RNA sample prep kit v2. cDNA libraries were prepared with 2,0 µg of RNA extracted from flash frozen floral buds. Co. spicatus whole flower library was sequenced using the Illumina HiSeq2000 at IIGB HT Sequencing Facility at the University of California, Riverside. All other samples were multiplexed 1:1 using barcode set A. Multiplexed libraries were sequenced using Illumina HiSeq2000 High Output at Vincent J. Coates Genomics Sequencing Lab at University of California at Berkeley. All libraries were sequenced as 100 bp pair-end reads.

Data cleanup and transcriptome assembly

Data clean-up was performed using a custom Perl script involving the following steps: (i) removal of identical forward and reverse reads; (ii) removal of duplicated reads in order to decrease the computational burden of subsequent de novo assembly; (iii) trimming of adapters, low complexity, and low quality (Q-score < 20) unique sequences using a combination of cutadapt v1.9.1 (Martin, 2011), Blat v348 (Kent, 2002), and Trimmomatic v0.35 (Bolger, Lohse & Usadel, 2014); (iv) screening of reads for contaminants against the human and Escherichia coli genomes using Bowtie v1.1.1 (Langmead et al., 2009). Clean-up quality was assed comparing FastQC v0.11.2 (http://www.bioinformatics.babraham.ac.uk/projects/fastqc/) reports of cleaned and raw reads.

Transcriptomes were assembled de novo using Trinity v2.1.0 (Grabherr et al., 2011) with a variety of parameters. The best assembly results (based on the quality assessments presented below) used default parameters for all other species despite discrepancies in the overall estimated transcriptome coverage and number of reads. Contigs larger than 300bp were retained and further annotated.

Quality assessment of de novo assemblies was performed using DETONATE v1.10 (Li et al., 2014). In particular, RSEM-EVAL was used as a reference-free evaluation method. True transcript length was estimated through comparison to several predicted transcriptomes from the sequenced genomes of Musa acuminata (D’Hont et al., 2012), the palms Phoenix dactylifera (Al-Mssallem et al., 2013) and Elaeis guineensis (Singh et al., 2013), and the core eudicot Arabidopsis thaliana (The Arabidopsis Genome Initiative, 2000). The number of coding sequences (CDS) in these species ranged from 28,889 in Phoenix dactylifera to 35,386 in Arabidopsis thaliana and 36,549 in Musa acuminata, to 44,360 in Elaeis guineensis.

Further quality assessment was performed on the basis of number and length of contigs as well as N50 (Table 2).

Table 2 Number of cleaned reads and contigs, average contig length in base pairs, and assembly quality metrics (N50 and RSEM-EVAL scores).

RSEM-scores for each transcriptome were calculated using Arabidopsis, Musa acuminata, Elaeis guineensis and Phoenix dactylifera predicted CDS as references.

Whole flower transcriptomes	Number of cleaned reads	Number of contigs	Average contig length	N50	RSEM-EVAL to Arabidopsis CDS	RSEM-EVAL to Musa CDS	RSEM-EVAL to Elaeis CDS	RSEM-EVAL to Phoenix CDS	
Musa bajsoo	6,103,473	59,607	1,177	1,635	−554.921.347	−554.925.496	−554.909.485	−554.930.293	
Orchidantha fimbriata	4,365,085	67,283	1,032	1,408	−396.133.340	−396.137.949	−396.118.692	−396.143.069	
Calathea zebrina	142,860,349	132,411	1,724	2,440	−994.730.221	−994.728.623	−994.727.315	−994.729.011	
Canna sp.	9,357,365	74,190	1,113	1,503	−860.726.519	−860.732.496	−860.711.867	−860.736.385	
Zingiber officinale	4,643,266	52,798	825	1,602	−357.355.187	−357.358.211	−357.346.889	−357.360.742	
Costus spicatus	1,292,595	19,377	632	674	−95.168.818	−95.169.800	−95.166.156	−95.170.392	
Dichorisandra thyrsiflora	6,252,788	64,723	891	1,166	−603.219.814	−603.224.077	−603.211.474	−603.225.657	

Transcriptome annotation and comparison

Statistically supported contigs were annotated with the help of TransDecoder v4.1.0 (https://transdecoder.github.io/). First, coding regions were identified using TransDecoder long ORFs prediction. Predicted long ORFs were subjected to a Blastp search (Gish & States, 1993) using Blast+ v2.7.1 against the Uniprot database (The UniProt Consortium, 2015), as well as a HMMER3 v3.1b2 (Eddy, 1998) search against the Pfam database (Finn et al., 2016). The results from the Blastp and HMMER3 searches were used by TransDecoder to filter likely coding regions from the predicted long ORFs list. For each species, TransDecoder-predicted coding regions were further filtered, using a Blastp search to the Uniprot database and the following parameters: ≥ 70% identity; E-value ≤ 1.0e−5; alignment length ≥ 100 bp; and coverage of at least 40%. These stringent lists were used as inputs for whole flower transcriptome comparisons, in order to avoid the inclusion in the analyses of chimeras and/or truncated transcripts.

Orthology between transcriptome predicted long-ORFs and CDS of sequenced genomes of Musa acuminata (D’Hont et al., 2012), Phoenix dactylifera (Al-Mssallem et al., 2013), Elaeis guineensis (Singh et al., 2013), and Arabidopsis thaliana (The Arabidopsis Genome Initiative, 2000) were established using OrthoFinder v2.2.3. Functional annotation of orthogroups were based on gene counterparts of the sequenced genomes of Arabidopsis thaliana (TAIR10) and Elaeis guineensis. Filtered contigs were also annotated based on nucleotide Blastn searches to predicted coding sequences (CDS) of the sequenced genomes listed above. Venn diagrams were built using Venny (Oliveros, 2007–2015), http://bioinfogp.cnb.csic.es/tools/venny/index.html), based on Elaeis guinensis Blastn results, especially in cases where no Arabidopsis counterpart was identified.

Transcription factor sorting and analysis

Transcripts were further classified into overall functional categories as either metabolic enzymes, mitochondrial, chloroplast, structural or regulatory, based on BLAST results. Unknown transcripts as well as predicted uncharacterized transcripts were grouped as “uncharacterized”. Regulatory transcripts were further analyzed regarding their role as transcription factors, and were subjected to further BLAST searches against the NCBI database, based on their conserved DNA-binding amino-acid domains. Further analysis also entailed a comparison of these transcripts to transcription factor sequences available at the curated plant specific database PlantTFDB v4.0 (http://planttfdb.cbi.pku.edu.cn/index.php; Jin et al., 2017). A list of all transcription factors retrieved in this analysis is presented on Supplemental Information S1.

All data processing was performed within the QB3 Computational Genomics Resource Laboratory (CGRL) at University of California at Berkeley, except when specified otherwise.

Results

Transcriptome assembly

The number of cleaned reads for each whole flower transcriptome ranged from ∼1 million reads for Co. spicatus to 142,860,349 reads in Ca. zebrina (Table 2). The significant difference in the number of reads is likely due to differences in the sequencing platform, in the case of Co. spicatus, and unequal multiplexing of libraries, in the case of Ca. zebrina. All other libraries resulted in a comparable number of reads, ranging form ∼4.3 million in O. fimbriata to ∼9.3 million reads in Canna sp. The number of non-filtered contigs ranged from ∼52,000 to ∼74,000, except in Co. spicatus (∼19,000) and Ca. zebrina (∼132,000), likely due to the discrepancy observed in the number of cleaned reads. With the exception of Co. spicatus, contig average length and N50 were comparable in all other libraries (Table 2). It is interesting to notice that, when compared to Z. officinale, a ∼35-fold increase in the number of reads in Ca. zebrina resulted in only a ∼2-fold increase in contig length and N50. With the exception of Co. spicatus and Ca. zebrina, all other species’ best assemblies resulted in values for number of contigs, N50 and average contig length (Table 2) comparable to those reported in the literature (e.g., 75 medicinal plant transcriptomes in Xiao et al., 2013; Stevia rebaudiana transcriptome in Chen et al., 2014; Musa acuminata root transcriptome in Zorrilla-Fontanesi et al., 2016).

In order to further assess assembly quality, we calculated RSEM-scores based on estimates of true transcriptome length of Musa acuminata, Phoenix dactylifera, Elaeis guineensis, and Arabidopsis thaliana (Table 2). Although we found no significant difference between results, RSEM-EVAL scores tended to favor the largest transcript length (Elaeis guineensis), regardless of phylogenetic proximity. Even for M. basjoo, phylogenetically close to Musa acuminata, the best RSEM-score was that based on Elaeis guineensis transcriptome.

Transcriptome annotation and comparison

Transcriptomes were filtered based not only on long predicted open reading frames (long ORFs) but also on Blastp and HMMER3 results (filtered ORFs) using TransDecoder (Table 3). The average number of filtered ORFs was ∼30,000, ranging from 13,122 in Co. spicatus to 55,360 in Ca. zebrina. The number of filtered coding sequences observed in this study is similar to already described numbers of floral unigenes of other non-model plants, which ranges between ∼25,000 (in buckwheat, Logacheca et al., 2011) to ∼80,000 (in Dendrocalamus latiflorus floral buds, Zhang et al., 2012). Whole flower transcriptome filtered ORFs represented on average 47% of reconstructed contigs, and ranged from 40 to 68%, similarly to what has been recently reported in Arabidopsis developing flowers (23,961 expressed genes; 67% of predicted CDS; Zhang et al., 2015). After filtering, the high number of contigs observed in Ca. zebrina was reduced to 55,360 ORFs, which is within the upper limits of already described non-model plant floral transcriptomes (see above).

Table 3 Number of predicted long open reading frames (ORFs) from TransDecoder.

Long ORFs were first predicted from the universe of de novo assembled contigs. Blastp and HMMER3 searchers were used to further filter long ORFs.

Whole flower transcriptomes	TransDecoder ORF predictions	
	Long ORFs	% contigs	Filtered ORFs	% contigs	
Musa basjoo	48,051	81	29,182	49	
Orchidantha fimbriata	39,003	58	26,790	40	
Calathea zebrina	85,437	65	55,360	42	
Canna sp.	43,932	59	29,366	40	
Zingiber officinale	39,214	74	24,463	46	
Costus spicatus	17,112	88	13,122	68	
Dichorisandra thyrsiflora	37,449	58	27,772	43	

In order to further annotate the contigs, OrthoFinder was used to establish orthology between the transcriptomes and the sequenced genome CDS. A total of 41,557 orthogroups were found (Supplemental Information S2), of which 17,418 had counterparts in at least one of the sequenced genomes included in the analysis. Over 24,000 groups had no CDS components in any of the analyzed genomes, which might suggest the persistance of chimeras and/or truncated ORFs within the filtered transcriptomes, Zingiberales specific genes, or a combination of the two. Arabidopsis thaliana CDS were present in 11, 511 orthogroups (Supplemental Information S3), while 5,907 orthogroups had no Arabidopsis counterparts but comprised other CDS from at least one of the other sequenced genomes. Orthogroup species overlap is presented on Table 4. Furthermore, OrthoFinder identified 6,916 orthogroups with all 10 species present. Of those, only 28 comprised single-copy orthogroups, in which one single ortholog was found for each especies (Supplemental Information S4).

Table 4 Orthogroup species overlap as predicted by OrthoFinder.

Largest number of orthogroup overlap per species is highlighted in bold. Ca. zebrina transcriptome shows the largest number of overlaps to all species, with the exception of Arabidopsis thaliana, potentially resulting from increased transcriptome coverage in that species.

SPECIES	A.  thaliana	C.  zebrina	Canna sp.	D.  thyrsiflora	E.  guineensis	M.  basjoo	M.  acuminata	O.  fimbriata	P.  dactylifera	Z.  officinale	
A. thaliana	11,511	10,403	10,298	10,049	10,814	10,089	10,543	9,161	9,627	9,448	
Ca. zebrina	10,403	29,032	20,338	15,927	11,405	19,778	12,072	16,904	11,212	16,879	
Canna sp.	10,298	20,338	25,460	14,822	11,225	18,149	11,726	15,503	10,830	15,524	
D. thyrsiflora	10,049	15,927	14,822	20,139	10,875	14,985	11,073	13,757	10,494	13,985	
E. guineensis	10,814	11,405	11,225	10,875	13,065	10,992	11,428	9,820	11,109	10,101	
M. basjoo	10,089	19,778	18,149	14,985	10,992	26,331	12,034	15,989	10,591	15,923	
M. acuminata	10,543	12,072	11,726	11,073	11,428	12,034	13,910	10,392	10,539	10,547	
O. fimbriata	9,161	16,904	15,503	13,757	9,820	15,989	10,392	22,244	9,644	14,164	
P. dactylifera	9,627	11,212	10,830	10,494	11,109	10,591	10,539	9,644	13,156	9,805	
Z. officinale	9,448	16,879	15,524	13,985	10,101	15,923	10,547	14,164	9,805	21,568	

Within Zingiberales transcriptomes, the largest orthogroup overlap was to the Musa acuminata genome, likely a reflection of their phylogenetic proximity. In all cases, Zingiberales transcriptomes largest orthogroup overlap to a non-Zingiberales genome was to Elaeis guineensis CDS.

One-hundred and forty-two (142) orthogroups were Arabidopsis thaliana-specific (Supplemental Information S5) with no counterparts in any of the other analyzed genomes. Given that all other genomes were from monocot species, this finding might reflect either Arabidopsis-specific or eudicot-specific genes. Further analyses are necessary to determine whether these genes are involved in eudicot- or Arabidopsis-specific flower development.

Blastn searches were conducted on the basis of Arabidopsis, Elaeis, Phoenix and Musa predicted CDS (Table 5). These searches produced variable results, potentially due to phylogenetic proximity and degree of genome sequence completeness. In general, all floral transcriptome Blastn searches resulted in a very small number of hits to Arabidopsis thaliana CDS, as expected due to its phylogenetic distance, indicating that although Arabidopsis is likely the best annotated plant genome to date, its phylogenetic distance to the study group makes fine-tuned statements of homology between Arabidopsis coding sequences and the predicted ORFs in the Zingiberales species studied here a challenging task. For instance, while 80.5% of Musa acuminata CDS were present amongst M. basjoo contigs, only ∼5% of Arabidopsis thaliana CDS were represented within the same assembly (Table 5), which is expected due to the nature of Blastn searches. Only a small number of Blastn hits were observed for Phoenix dactylifera, likely indicating incompleteness of the current genome sequence: ∼29% of D. thyrsiflora contigs matched Elaeis guineensis CDS, while the same contigs matched only ∼19% of Phoenix dactylifera CDS (Table 5). In order to avoid phylogenetic bias, as well as to maximize transcriptome annotation, further Blastn analyses of filtered ORFs were based on Elaeis guineensis predicted CDS.

Table 5 Blastn results between floral transcriptomes and predicted coding sequences (CDS) from the genomes of Arabidopsis thaliana, Musa acuminata, Phoenix dactylifera, and Elaeis guineensis.

Transcriptomes	Musa acuminata	Elaeis guineensis	
Blastn all contigs to CDS	CDS represented in transcriptome	% CDS represented in transcriptome	Blastn all contigs to CDS	CDS represented in transcriptome	% CDS represented in transcriptome	
Musa bajsoo	49,127	29,433	80.5	19,509	19,945	44.96	
Orchidantha fimbriata	38,170	20,289	55.5	21,317	18,238	41.11	
Calathea zebrina	75,885	20,671	56.5	42,638	17,229	38.84	
Canna sp.	35,597	20,522	56.1	19,353	17,723	39.95	
Zingiber officinale	16,901	14,322	39.2	8,725	11,886	26.79	
Costus spicatus	9,319	9,223	25.2	4,491	6,430	14.5	
Dichorisandra thyrsiflora	12,384	8,596	23.5	11,394	12,780	28.81	
Transcriptomes	Phoenix dactylifera	Arabidopsis thaliana	
	Blastn all contigs to CDS	CDS represented in transcriptome	% CDS represented in transcriptome	Blastn all contigs to CDS	CDS represented in transcriptome	% CDS represented in transcriptome	
Musa bajsoo	15,586	9.015	31.21	2,136	1,571	4.44	
Orchidantha fimbriata	17,473	8,185	28.33	2,268	1,410	3.98	
Calathea zebrina	35,226	7,685	26.6	5,108	1,591	4.5	
Canna sp.	14,940	7,702	26.66	2,055	1,624	4.59	
Zingiber officinale	6,544	5,077	17.57	1,436	1,295	3.66	
Costus spicatus	3,354	2,816	9.75	706	743	2.10	
Dichorisandra thyrsiflora	8,856	5,403	18.7	1,827	1,507	4.26	

Based on Blastn searches against Elaeis guineensis predicted CDS, floral transcriptomes shared 4,153 genes (Fig. 2). We also identified 1,748 hits specific to Zingiberales, 666 to the ginger clade, 1,560 hits unique to the Cannaceae-Marantaceae lineage, 2,001 specific to the banana families, and 1,887 specific to Z. officinale, from which 221 hits are shared with Co. spicatus. The small number of contigs recovered for Co. spicatus likely limited the analysis of the Costaceae-Zingiberaceae lineage-specific Blastn hits (Supplemental Information S6).

Figure 2 Venn diagram of Blastn results of all floral transcriptomes filtered ORFs against Elaeis guineensis predicted CDS.

Values represent number of unigenes.

Conserved genes

Orthogroup analysis containing Arabidopsis thaliana counterparts (Supplemental Information S3) revealed the presence of several well-known gene families in our flower transcriptomes. Within these orthogroups, the most noticeable groups were members of the AGAMOUS-like (AGL) family of transcription factors, including AGL6, AGL12, AGL20, AGL26, AGL29, AGL44, AGL58, AGL61, AGL65 and AGL104. Other MADS-box genes, widely implicated in floral organ identity, were also identified such as APETALA3 (AP3), PISTALLATA (PI), and SEPALLATA3 (SEP3). Other MADS-box gene families involved in flower and fruit development were represented within the orthogroups: CAULIFLOWER (CAL), SHATTERPROOF2 (SHP2), CRABS CLAW (CRC), SHORT VEGETATIVE PHASE (SVP), TRANSPARENT TESTA16 (TT16), FLOR1 (FLR1), BELL1 (BEL1), as well as several members of the TCP/TEOSINTE BRANCHED family (TCP1, TCP3, TCP12, TCP15, and TCP24). Orthogroups lacking Arabidopsis thaliana counterparts further reinforced the presence of AGAMOUS-like genes, such as AGL61, AGL62 (three orthogroups), AGL80, as well as MADS32 (O’Maoileidigh, Graciet & Wellmer, 2014).

Blastn hits to E. guineensis were used to further place genes in functional categories, as described in methods. Figure 3 depicts the main category of genes shared by all floral transcriptomes. Almost half of these genes (47%) are enzymes related to metabolic processes of the cell, while 26% of the genes are structural proteins such as membrane proteins, cytoskeleton-related proteins, ribosomal, histones, heat-shock and ribonucleoproteins. Approximately 10% of these genes are regulatory proteins, of which approximately 508 could be assigned to known transcription factor (TF) families, based on the PlantTFDB v4.0 (Supplemental Information S1). From the 58 well-characterized plant transcription factor families, our dataset was able to retrieve 36 families, based on the closest homolog in Arabidopsis thaliana (Table 6).

Figure 3 Distribution of main ‘functional’ categories of coding genes shared by all floral transcriptomes, and shared by all Zingiberales floral transcriptomes based on Blastn results to Elaeis guineensis transcriptome.

Table 6 Distribution of transcription factor families amongst the floral transcriptomes studied.

A total of 508 transcription factors were ascribed to 36 of the 58 plant transcription factor families characterized in the PlantTFDB v4.0. Outgroup species is D. thyrsiflora.

	Shared by all	Zingiberales	Banana clade	Ginger clade	Canna-Calathea	Zingiber	Outgroup (Dichorisandra thyrsiflora)	
Transcription Factor Families (PlantTFDB v4.0)	25	22	19	18	20	30	21	
Putative Transcription Factors (not in PlantTFDB v4.0)	0	0	1	0	2	3	2	

Additionally, six putative new categories of TFs that are not described in the database were also recovered, although more experimental evidence is required to further categorize their potential role as transcription factors. Here, we preliminarily named these sequences based on their match to the NCBI Conserved Domain Dataset (https://www.ncbi.nlm.nih.gov/cdd): Bromodomain-family (five unique sequences: GTE4-like, GTE6-like, and GTE9-like homologs in the Canna-Calathea clade; GTE7-like homologs in Z. officinale; and GTE9-like homologs in Dichorisandra thyrsiflora); PUR-A family (one unique sequence: PURA1-like homolog in the Canna-Calathea clade); YL1 domain family (1 unique sequence: SWR1 complex subunit 2-like homolog in D. thyrsiflora); TFIIS-domain family (one unique sequence: IWS1-like homolog in Z. officinale); LIM-domain family (two unique sequences: SEUSS-like homologs in the banana clade) and SAND-domain family (one unique sequence: UTLRAPETALA1-like homolog in Z. officinale) (see Supplemental Information S1).

Interestingly, the remaining regulatory proteins that were not included in the transcription factor category were nonetheless implicated in regulating plant organ development and/or growth, acting as protein co-factors that physically interact with transcription factors, or as related to the chromatin remodeling machinery.

Among the transcription factors shared by all flower transcriptomes, it is worth noticing a single homolog of APETALA-2 (a member of the A-class ABC model genes Jofuku et al., 1994), three homologs of MADS-6 or AGL6, as well as several homologs of HUA2-like proteins 2 and 3. In Arabidopsis thaliana, HUA1 and HUA2 are important components of the AGAMOUS gene regulation pathway (Chen & Meyerowitz, 1999). It has been suggested that HUA2 facilitates AGAMOUS action during flower development (Chen & Meyerowitz, 1999), and it is also required for the expression of FLC in Arabidopsis thaliana (Doyle et al., 2005). Moreover, HUA2 has been implicated in natural variation in Arabidopsis thaliana shoot morphology (Wang et al., 2007). Five LEUNIG-like homologs were also recovered in all floral transcriptomes. LEUNIG proteins are also involved in the regulation of AGAMOUS expression in Arabidopsis thaliana (Liu & Meyerowitz, 1995; Sridhar et al., 2004). The number of shared genes involved in the regulation of AGAMOUS indicates the shared importance of precise AGAMOUS regulation during flower development (Supplemental Information S1). In particular, genes involved in physiological responses to stress and pathogen response, such as the WRKY family of transcription factors (Wang et al., 2011) and the NAC domain proteins (Nuruzzaman, Sharoni & Kikuchi, 2013), were recovered in all transcriptomes. More recently, WRKY71 has been implicated in the control of shoot branching in Arabidopsis thaliana, through the regulation of RAX genes (Guo et al., 2015). All floral transcriptomes also presented several members of the zinc-finger transcription factor family, seven KNOTTED 1-like homologs, as well as GATA transcription factors 2, 4, 12, and 24. Several members of the bHLH family; homologs of MYB44, MYB82; TCP-4, -15, and -7 homologs; four CONSTANS-like homologs; several members of the TCP family, as well as WUSHEL-like transcripts were also widely retrieved (Supplemental Information S1).

Other regulatory proteins include, for example, a homolog of COBRA-like 1; two homologs of FY-like proteins; one FRIGIDA-like homolog; and five homologs of EMBRYONIC FLOWER2-like. We also retrieved six TOPLESS-like homologs, almost 20 members of the TBC1 family, five IWS1 homologs, a GIGANTEA-like homolog, as well as four SQUAMOSA PROMOTER BINDING-like homologs.

Interestingly, the most prominent feature of the Blastn searchers was the match to different paralogues and/or variants of the same genes or gene families in different floral transcriptomes (Supplemental Information S1). For example, LATERAL ORGAN BOUNDARIES (LOB)-domain homologs were retrieved in all floral transcriptomes analyzed. However, while LOB40, 41 and 6-like homologs were retrieved in all Zingiberales floral transcriptomes, LOB36 and a paralog of LOB6-like transcripts were retrieved only in the banana transcriptomes. Similarly, LOB18-like was only recovered in the Cannaceae-Marantaceae lineage, while LOB4-like transcript was only recovered in Z. officinale. On the other hand, LOB15-like homologs were only recovered in the floral transcriptome of D. thyrsiflora. LOB genes have been implicated in defining organ boundaries in Arabidopsis floral organs through negative regulation of brassinosteriod accumulation (Shuai, Reynaga-Peña & Springer, 2002; Bell et al., 2012).

Whether this phenomenon is a result of gene duplication followed by divergence or whether it is due to lineage-specific divergence within a single copy begs further investigations. Whether these homologs have retained the same function is an exciting matter for further studies.

Lineage-specific genes

The great majority of lineage-specific genes, including Zingiberales specific genes, were related to metabolic processes of the cells (Fig. 3). The most prevalent unique genes were enzymes such as oxidoreductases, methyltransferases, aminoacyl-tRNA synthetases, kinases, hydrolases, and phosphatases. Carrier proteins, transporters, chaperones and ribonucleoproteins were also abundant in all lineage-specific datasets. Several transcription factors, many of which are known players during plant development, were recovered in a lineage-specific fashion. Fifty percent of Zingiberales specific genes are metabolic enzymes (28%) or structural proteins (22%), while 12%, approximately 210 coding sequences, are regulatory proteins (Fig. 3).

Among these regulatory proteins, several families of transcription factors were recovered exclusively in the Zingiberales, such as ENHANCER OF AG-4, various AP2-like ethylene-response transcription factors, BRZ1 homologs 1 and 3, SHOOT GRAVITROPISM 5-like homolog, the zinc-finger transcription factor JACKDAW-like homolog, a YABBY2-like homolog, as well as GT-2 and GT-3 (GT-element binding transcription factors) homologs.

Several DIVARICATA lineage-specific homologs, were retrieved in the banana and ginger groups transcriptomes. Similarly, other homologs appeared in a lineage-specific manner. For example, while two homologs of B-ZIP transcription factor family TGA4-like were recovered in the banana group, homologs for TGA2-like were recovered only in the ginger clade. Likewise, homologs of the trihelix DNA binding family gene ASIL1-like (ARABIDOPSIS 6B-INTERACTING PROTEIN 1-LIKE) were recovered in the banana group, while ASIL2-like homologs were recovered in the ginger clade.

As far as other regulators go, in all Zingiberales floral transcriptomes, but not in the outgroup D. thyrsiflora, we were able to recover a homologue of the plant homeodomain (PHD) protein ING2 (Inhibitor of growth). ING tumor suppressors are found in animals, plants and yeast, and have long been implicated in oncogenesis, control of DNA damage repair, cellular senescence and apoptosis (Champagne & Kutateladze, 2009). In A. thaliana, ING2 is involved in chromatin regulation by binding to the active histone marker H3K4me3/2 (Lee et al., 2009). Histone modifications, such as those promoted by ING2 and other PHD proteins, modulate the expression of crucial genes involved in flower development (López-González et al., 2014). Similarly, the histone chaperone ANTI-SILENCING FACTOR-1 (ASF1) homologue was recovered in all analyzed Zingiberales transcriptomes, while missing in D. thyrsiflora. ASF1 is a family of histone chaperones conserved in all eukaryotes (Triphathi et al., 2015), and in A. thaliana ASF1 is required for cell proliferation during development and is involved in transcriptional regulation of histones and histone modifications (Zhu et al., 2011). However interesting, further analyses are necessary to establish the potential role of histone modifications, and in particular the functions of ING2 and ASF1, in Zingiberales flower development.

In turn, various transcription factors were only recovered in the D. thyrsiflora floral transcriptome to the exclusion of the Zingiberales. Among these are a FLORICAULA/LEAFY homolog, a homolog of ODORANT1-like, a homolog of JUNGBRUNNEN 1-like, homologs of RAX- 1, -2, and -3, as well as homologs of the transcription factors DPB, TT2-like, and GAMYB-like. In particular, a SOMBRERO- like homolog was retrieved only in D. thyrsiflora. SOMBRERO proteins, members of the NAC domain transcription factors, have been implicated in the control of cell division plane orientation in Arabidopsis thaliana Willemsen et al., 2008. Other regulators retrieved specifically in the Dichorisandra lineage include two STICHEL-like homologs, a homolog of UPSTREAM OF FLC-like, a TONSOKU-like homolog, two SAGA-like homologs, a TASSELSEED homolog, and a TITAN-like homolog.

Regulatory sequences retrieved exclusively within the banana lineage, represented by M. basjoo and O. fimbriata floral transcriptomes, include four CCA1-like homologs, six FLX2-like homologs, a KTI12-like homolog, a YABBY4-like homolog, a CPC homolog, and a SPATULA homolog represent transcription factors that were recovered exclusively in this group. Curiously, few coding sequences were uniquely reconstructed within the ginger clade, potentially due to the low coverage of the Co. spicatus transcriptome. Particularly interesting is the unique recovery of four AS1-like (ASYMMETRIC LEAVES-1) homologs and two DROOPING LEAF-like genes. Regulatory coding sequences uniquely reconstructed in the Canna-Calathea (Cannaceae-Marantaceae) lineage include a CUC2 homolog, a homolog of Arabidopsis EXORDIUM-like protein, two FAF-like homologs, and five SPX-like homologs.

A complete list of lineage specific transcription factors, sorted by plant transcription factor families characterized in the PlantTFDB, can be found in Supplemental Information S1.

Discussion

Recently, there has been an explosion in the use of RNA-Seq approaches as part of a comparative analysis pipeline to study the evolution of developmental processes, using plant transcriptomes as an indication of differential gene expression among organisms with different phenotypic displays. This approach has become particularly important in non-model organisms that lack a reference genome or other genetic and bioinformatic tools that exist in plant model organisms like A. thaliana, rice, poplar or corn. Despite challenges assembling transcriptomic sequence data without a reference genome, researchers can determine the quality of their data based on the number, size and scores of the contigs assembled. The transcriptome data presented here are in agreement in terms of number of contigs, contig size distributions, and quality scores with those presented in the literature.

The study of mechanisms underlying floral diversification in plant lineages will likely point, in most cases, to at least three potentially concurrent scenarios: (i) tinkering of conserved mechanisms specific to flower development; (b) evolution of lineage-specific mechanisms resulting in novelty or change, or (c) co-option of non-flower mechanisms to elaborate specific aspects of flower development. The identification of these mechanisms, however, requires careful examination of exemplar species within a clearly delimited phylogenetic context. Also, careful choice of outgroup species might help the distinction between gain versus loss of molecular processes when analyzing lineage specific phenomena. Our data show that the inclusion of D. thyrsiflora significantly reduced the overall number of Zingiberales unique genes, as well as the number of lineage specific genes within the Zingiberales, potentially due to shared molecular mechanism during flower development. It is possible that the addition of other outgroups would further limit the lineage-specific datasets. The results presented here support previous assertions that annotation based on Blastn searches is highly influenced by phylogenetic proximity as well as genome sequence completeness and annotation quality, particularly when blasting against predicted CDS (Hornett & Wheat, 2012). Meanwhile, orthogroup analysis provides a wider view of less stringent relationships between trasncriptomes. Furthermore, the orthogroup analysis presented here reinforces the notion that gene duplications are a widespread phenomenon during plant evolution (Panchy, Lehti-Shiu & Shiu, 2016). Only 28 of the over 40,000 orthogroups identified comprised single copy genes in the transcriptomes and genomes analyzed.

The stringent filtering of the data performed with Blastn likely excluded several genes that could potentially participate in flower development across the Zingiberales and in the outgroup (D. thyrsiflora), and may even participate in floral evolution. However, due to this stringent cutoff, it is likely that the genes recovered are strong candidates for further studies. Functional analysis of the genes that emerge from these comparative datasets, coupled with careful phylogenetic assessments of specific gene families, will potentially refine the picture.

Perhaps the most significant results presented here are related to the set of shared floral transcription factors recovered for all taxa analyzed. Due to the nature of the methodology used, we believe there is sufficient evidence to support the presence of these genes in all floral transcriptome studies, making them likely floral development regulators and involved in not only floral development but, given their presence among and between lineages, suggesting that they are conserved regulators of floral evolution. Most of these genes and gene families have already been implicated in floral development in A. thaliana, but knowledge of their roles outside core eudicots is still poor. Their specific involvement in processes of morphological diversification has yet to be established.

Our results point to interesting differences between Zingiberales lineages. In particular, the presence of a YABBY4-like homolog in the banana lineages but not in the ginger clade—where only a YABBY2-like homolog was reconstructed—might underlie developmental differences between these Zingiberales flowers. Information regarding the role of YABBY4 in comparative floral development remains sparse. Even though expression of YABBY4 (INNER NO OUTER) is restricted to the ovule integument (Villanueva et al., 1999) and seems to be conserved across angiosperms (Skinner et al., 2016), little is known about the presence of this gene in monocots other than rice, or pertaining the role it may play in ovule development within the monocot clade (Toriba et al., 2007; Morioka et al., 2015). Although it requires further evidence, the lineage specific gene set presented here might provide an interesting candidate gene list for further studies into the molecular mechanisms of floral development and diversification in the Zingiberales.

It is widely accepted that the ability to recover low expressed genes is related to transcriptome coverage (Grabherr et al., 2011; Martin & Wang, 2011; Tarazona et al., 2011). The high coverage of Calathea might explain the large number of genes recovered that appear unique to Cannaceae-Marantaceae, especially given the overrepresentation of transcription factors in this lineage. However, the total number of unique transcription factors between Canna and Calathea is similar to that observed in other lineages within the Zingiberales. Particularly interesting was the reconstruction of CUP-SHAPED COTYLEDON2 (CUC2) exclusively in the Cannaceae-Marantaceae lineage. The evolution and functional divergence of CUC genes (1–3) have been well studied in Arabidopsis (Hasson et al., 2011), although much less is known in monocots especially outside of the grasses. During flower development, CUC genes have been implicated in the formation of carpel margin meristems, although their role in plant development does not appear to be restricted to the flower (Kamiuchi et al., 2014). It is conceivable that the CUC gene copies play important roles, together with SPATULA homologs (SPT) (Nahar et al., 2012), in carpel diversification in Zingiberales.

It is interesting to notice that AGAMOUS regulatory proteins were widely recovered in all transcriptomes, suggesting consistent levels of expression throughout the Zingiberales and outgroup developing flowers. This might support the evolution of several regulatory mechanisms of AGAMOUS expression during flower development, bringing redundancy and indicating the critical nature of AGAMOUS regulation. In turn, it may suggest that variations of AGAMOUS expression might lead to floral morphological diversification, a mechanism that has already been proposed to participate in Zingiberales flower evolution (Almeida et al., 2015b).

Because expression levels can interfere with the ability to reconstruct specific genes, it is possible that some of the differences observed in lineage-specific transcriptome reconstructions, particularly the absence of transcripts, are due to low or restricted expression within the developing flower. It is imperative that further studies are carried out, especially comparative spatial–temporal expression studies, to further unravel the role of these transcription factors in floral morphological variation. Comparisons based on expression levels of shared genes, as well as protein–protein or protein-DNA interactions, can certainly reveal other levels of developmental divergence. Expression levels were not calculated here, due to the lack of replicates for each floral transcriptome. Also, despite the interesting findings discussed here regarding coding sequences and, in particular transcription factors, further analysis is needed to fully uncover the mechanisms underlying floral developmental evolution. A careful analysis of non-coding sequences might revel other layers of gene regulation and function that were not explored in this work. The complexity of the molecular mechanisms underlying floral development cannot be underestimated. Thus, we believe that further investigations are needed to achieve a full understanding of the molecular processes underlying flower developmental evolution in the Zingiberales.

Despite limitations, we believe the transcriptome analysis presented here sheds light on interesting phenomena that might underlie molecular mechanisms of flower developmental evolution. In particular, the consistent recovery of distinct homologs for various genes families in closely related evolutionary lineages is a pattern that suggests the need for further studies. The complex patterns of gene duplications in plants, although daunting, provides an exciting opportunity for the study of the relationship between genes, functions and morphological diversification.

Supplemental Information

Supplemental Information 1 List of all Transcription Factors retrieved in this analysis

Click here for additional data file.

Supplemental Information 2 List of all orthogroups from OrthoFInder

Click here for additional data file.

Supplemental Information 3 List of all orthogroups with at least one Arabidopsis thaliana counterpart

Click here for additional data file.

Supplemental Information 4 Single-copy orthogroups

Click here for additional data file.

Supplemental Information 5 Arabidopsis thaliana specific orthogroups

Click here for additional data file.

Supplemental Information 6 Costaceae-Zingiberaceae lineage specific Blastn hits

Click here for additional data file.

The authors would like to acknowledge H.Forbes from UC Botanical Garden, and E.Sandoval from UC Davis Greenhouses for support with plant materials; K.Bi from qB3 Computational Genomics Lab (UC Berkeley) for help with data processing and L.Smith from the Evolutionary Genetics Lab (UCB) for support during library preparations.

Additional Information and Declarations ]

Competing Interests

Author Contributions

DNA Deposition

The authors declare there are no competing interests.

Ana Maria R. Almeida conceived and designed the experiments, performed the experiments, analyzed the data, contributed reagents/materials/analysis tools, prepared figures and/or tables, authored or reviewed drafts of the paper, approved the final draft.

Alma Piñeyro-Nelson conceived and designed the experiments, performed the experiments, analyzed the data, prepared figures and/or tables, authored or reviewed drafts of the paper.

Roxana B. Yockteng conceived and designed the experiments, performed the experiments, authored or reviewed drafts of the paper.

Chelsea D. Specht conceived and designed the experiments, contributed reagents/materials/analysis tools, authored or reviewed drafts of the paper.

The following information was supplied regarding the deposition of DNA sequences:

Sequence Read Archive experiment accession: SRX2920013

Experimental raw data associated with project:

SRR5685225–SRR5685230.

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
