# Peer review of "Comparative analysis of whole flower transcriptomes in the Zingiberales"

_PeerJ, doi:10.7717/peerj.5490_

## Round 0.1 · original submission · Minor Revisions

Dear Ana Maria, Thanks again for sending your paper to PeerJ. I'm happy to inform you that reviewers suggested that your paper might be appropriate for publication in PeerJ pending minor corrections. With best regards,

Yves Van de Peer

·

Basic reporting

no comment

Experimental design

no comment

Validity of the findings

no comment

Additional comments

Almeida et al. assembled floral transcriptomes for seven linages in the Zingiberales, revealing unique gene conservation and duplication patterns which may be related to divergent and unique floral morphology within the family. These transcriptome assemblies will be useful for the comparative genomics and floral development research communities. I have a few issues that I feel should be addressed to strengthen this manuscript prior to publication.

Major issues:
1. Analyses related to orthology and conserved genes are potentially problematic.
Blastp or blastx would return more hits than blastn, especially in divergent taxa. This explains why only 2-5% of transcripts from the Zingiberales BLAST to Arabidopsis orthologs; the nucleotide divergence between monocots and eudicots is too high. Hits to Elaeis are also probably lower than expected given it’s phylogenetic placement. Using blastp or blastx would help classify more proteins which could affect the interpretation of gene conservation and lineage specific changes.
A more robust way to establish orthology would be to use orthoMCL or orthofinder. The resulting orthogroups can then be assigned functional categories based on pfam domains or Arabidopsis homology.

2. More analyses related to gene conservation/ lineage specific changes would be interesting. BLAST results are capturing only a rudimentary survey of gene-level changes. Analyses through a pylogenetic framework would be useful to identify gene duplications and ka/ks divergence related to adaptive evolution of floral morphology. Such analyses may me beyond the scope of this manuscript.

Minor:
Line 122 on. The build/version for each bioinformatic program should be provided.

Reviewer 2 ·

Basic reporting

No comment.

Experimental design

Obviously replication of the experiments would be desirable but that is not provided. Nonetheless the resources described here would be beneficial to the community.

Validity of the findings

No comment

Additional comments

The authors have developed transcriptomics datasets to compare the gene expression profiles of members of the Zingiberales order. The goal was to shed light on the evolution and development of these flowers. The task is at hand is extremely difficult and while the data presented is limited, it provides a decent basis for further work. There are a few things that I would recommend changing and have given some suggestions regarding analysis or discussion that could be included.

1. There are a lot of numbers thrown about in the manuscript, which reduces the readability of the manuscript. I would recommend reducing the prevalence of this in the text (although some are absolutely necessary). For example, “The number of filtered coding sequences observed in this study is similar to already described numbers of floral unigenes of other non-model plants…”, which goes on to list many numbers. To improve the readability I would keep the sentence in quotes above, remove the rest and make a table or simply give the upper lower limit examples while keeping the references.
2. Ln 102: “whole developing flowers of…”
- Could the authors be more specific here? It would be nice to know if flowers at several different stages were used. What morphological markers guided the collection strategy?
3. Ln 92: “we present a comprehensive analysis…”
- This wording is a bit strong.
4. Ln 293: I don’t understand why Table 5 is referred to here when discussing chromatin remodeling proteins.
5. I missed a discussion on the relevance of the floral homeotic protein activity during the manuscript, although the upstream activation of AG was discussed. Perhaps it would be good to include how many of the genes identified as commonly expressed and “differentially” expressed are putative or likely targets of the homeotic proteins as there is a considerable amount of work done on this in Arabidopsis (Kaufmann 2009; Kaufmann 2010; Pajoro 2014; O’Maoileidigh 2013; Wuest 2012). I’m not insisting on a new analysis but a discussion of how the activity of these proteins might be important (or not) during the development of Zingiberales flowers might improve the context of some of the findings.
6. Similar to point 5, some work has been on the stage-specific expression of genes during Arabidopsis flower development. The authors could look into these data as a comparative resource (Wellmer 2006; O’Maoileidigh 2015; Jiao 2010). Again I’m not insisting on a new analysis but it is something the authors can consider for now or later.

---

## Round 0.2 · accepted · Accept

The authors have done their utmost best to address all comments of the reviewers and this paper can now be accepted for publication in PeerJ.

#